# Corn: Its Structure, Polymer, Fiber, Composite, Properties, and Applications

**DOI:** 10.3390/polym14204396

**Published:** 2022-10-18

**Authors:** Abdulrahman A. B. A. Mohammed, Zaimah Hasan, Abdoulhdi A. Borhana Omran, V.Vinod Kumar, Abdulhafid M. Elfaghi, R. A. Ilyas, S. M. Sapuan

**Affiliations:** 1Institute of Sustainable Energy, Universiti Tenaga Nasional, Jalan Ikram-Uniten, Kajang 43000, Malaysia; 2Department of Mechanical and Mechatronic Engineering, Faculty of Engineering, Sohar University, Sohar P C-311, Oman; 3Department of Mechanical Engineering, College of Engineering Science & Technology, Sebha University, Sabha 00218, Libya; 4Faculty of Mechanical and Manufacturing Engineering, Universiti Tun Hussein Onn Malaysia, Batu Pahat 86400, Malaysia; 5Faculty of Chemical and Energy Engineering, Universiti Teknologi Malaysia, Johor Bahru 81310, Malaysia; 6Advanced Engineering Materials and Composites Research Center (AEMC), Faculty of Engineering, Universiti Putra Malaysia, Serdang 43400, Malaysia

**Keywords:** corn starch, corn fiber, corn biocomposite, properties improvement

## Abstract

Biocomposite materials have a significant function in saving the environment by replacing artificial plastic materials with natural substances. They have been enrolled in many applications, such as housing, automotive engine components, aerospace and military products, electronic and circuit board components, and oil and gas equipment. Therefore, continuous studies have been employed to improve their mechanical, thermal, physical properties. In this research, we conduct a comprehensive review about corn fiber and corn starch-based biocomposite. The results gained from previous studies were compared and discussed. Firstly, the chemical, thermal, and mechanical properties of cornstarch-based composite were discussed. Then, the effects of various types of plasticizers on the flexibility of the cornstarch-based composite were addressed. The effects of chemical treatments on the properties of biocomposite using different cross-linking agents were discussed. The corn fiber surface treatment to enhance interfacial adhesion between natural fiber and polymeric matrix also were addressed. Finally, morphological characterization, crystallinity degree, and measurement of vapor permeability, degradation, and uptake of water were discussed. The mechanical, thermal, and water resistance properties of corn starch and fibers-based biopolymers show a significant improvement through plasticizing, chemical treatment, grafting, and cross-linker agent procedures, which expands their potential applications.

## 1. Introduction

After the industrial revolution, manufacturing processes used modern systems and started to cause a huge damage to the environment by increasing carbon emission, global warming, and producing more non-degradable materials. All these issues affect the life cycle of many species on the earth. Because of all these problems, scientists applied many solutions to reduce environmental pollution. Thus, in order to overcome these problems, scientists had developed the ecofriendly engineering materials, which are degradable, low in manufacturing cost, light weight, renewable, chemically inert, and easy to process compared to most artificial plastic materials [1,2,3,4].

Developing the properties of biocomposite material is still an ongoing process. Numerous investigations are being conducted on plants, both wood and non-wood, to extract fibers for biocomposite materials. The ingredients of biocomposite materials are extracted from various types of agricultural crops residue, such as wheat, corn, cassava, hemp, jute, kenaf and other crops [5]. The factors that make plants more useful than other sources are availability, quality, quantity, physical, and chemical properties, such as cellulose content in fiber, and degree of polymerization [6]. Furthermore, the degree of crystallinity, density, and porosity makes a difference in the production process. In addition, the starch that is extracted from natural resources makes a good alternative even for artificial polymers.

There is inherently low compatibility between natural fiber and the hydrophobic thermoplastic matrix because the natural filler is naturally hydrophilic. This incompatibility between the natural filler and thermoplastic causes problems in composite processing and material properties. In order to overcome these incompatibility problems, various physical and chemical methods have been employed to modify the natural filler [7]. Therefore, natural fiber modifications were considered in modifying the fiber surface properties to improve their adhesion with different ways and this can be done physically or chemically [8,9]. Plenty of studies discussed the characterization and properties of fiber to achieve the best results using several surface modification methods. These methods include different chemical treatments, such as alkali, saline, alkali-saline, compatibilizers, and adding cross-linking agents, such as glutaraldehyde (GLA) and epichlorohydrin (EP). Physical methods include stretching and thermo-treatment. Physical treatments do not extensively change the chemical composition of the fibers. Therefore, the interfacial binding between the fiber and the matrix is generally enhanced due to chemical bonding that created between the fiber and the matrix. Adding plasticizers improve the properties of the material by increasing the flexibility of the material. There are many types of plasticizer such as, fructose, sorbitol, urea and glycerol [8,10], while the main plasticizer is usually the water because of its capacity to hydrolyze the molecular link structure of starch when heated together [11].

Corn is one of the most cultivated crops globally, whereby according to United States Department of Agriculture the worldwide production reached 1110.84 million metric tons in 2019 [12]. It is a non-wood plant [13,14], which is planted in many countries and have a lot of residues. Corn can be considered a rich source of both fiber and starch. Corn starch and corn hull fiber can be extracted from the granules by the wet milling process. Corn stalk (CS) and husk can be obtained from the stems and leaves by drying, grinding and sieving [13].

Percentages of corn in the biocomposite significantly differ for each composition depending on the material properties used as matrix and chemical bonding created between the filler and the matrix. To achieve perfect mechanical, thermal, and chemical properties, selecting the right composition needs to be determined experimentally. Usually, after a specific point or percentage of corn load, the properties drop, which requires chemical or thermal treatment or adding other materials to defeat the drop. Corn has been mixed with other materials to enhance their properties, such as recycled low density polyethylene (RLDPE) [15], calcium carbonite CaCO_3_ [16], microalgae [17], poly(lactic acid) [18], chitosan [19,20,21,22], and polypropylene [23].

This paper aims to make a comprehensive review on the previously established studies regarding preparation, characterization, and properties of different parts of the corn plant. In addition, the paper discusses the effect of fiber treatment and the addition of cross-linking and plasticizers to corn starch-based biocomposites for different applications.

## 2. Corn Plant

The changes in the social structures in the 18th and 19th centuries led to high demand in biocomposites [24]. In 1907, Leo Baekeland invented Bakelite the first fully synthetic plastic [25]. After that, numerous of plastic application invented and cause a tremendous consumption of plastic, which negatively affected the environment as a non-degradable material that creates a necessity to use degradable material rather than fossil fuels and plastic.

From ancient times, builders, artisans, engineers, and manufacturers continued to develop composites from a wider array of materials for more sophisticated applications. The concept of combining different materials in the building has appeared in 3400 B.C.E [26].

Many types of natural fiber have been studied as a potential replacement of artificial fiber, such as glass and carbon fiber [27]. The sustainability of the crops and plants as source of biocomposite material forms encourages scientists to develop more biocomposite. In recent years, there are many types of natural fiber based plants. Figure 1 illustrates the classification of natural fiber.

### 2.1. Historical Background of Corn

Corn cultivation was the main reason for many communities towards civilization, farming, and living in one place. Archeological data have shown that corn was cultivated by 2000–2500 B.C.E [29]. Since then, corn has been used as source of food for both human being and animals.

In 2014, 36 million hectares of corn were planted in the United States of America (USA) [30]. The production reached 347.782 million metric tons at USA in 2019 [12]. More tonnage of corn is produced worldwide each year than any other major crops. From 2005 to 2007, the world corn productions were 736 million metric tons, surpassing wheat and rice by 122 million and 92 million metric tons. That gap is projected to continue growing in the future [30]. The main parts of the corn plant are the roots from the bottom, stalk, husk, silks, ear of corn, and tassel from the top [31,32,33].

### 2.2. Composition of Corn Plant

The structure of the corn kernel consists of six physical parts, which can be identified as the tip cap, the hull (source of fiber [34,35]), the horny gluten, the horny starch, the white starch, the germ [36]. Figure 2 shows parts of corn plant. Table 1 and Table 2 show the chemical composition and physical properties of corn starch, corn stalk, corn hull, corn husk and corn cob. The other parts of the plant can be considered as a source of fiber, the most important source of fiber are stalk [37], husk [38], and corn cob [39].

## 3. Starch Extraction and Preparation

The process of extracting corn starch can be done by one of the following methods:-Starch extracted after steeping for 24–72 h at 45 °C using centrifugation or sedimentation separation method [40].-Starch was extracted after steeping for 20–24 h at 50 °C using the centrifugation method [41];-Starch extracted after steeping for 30 min using the decantation method [42].

The corn grain composition consists of starch 68.0%–74.0 %, hemicellulose and cellulose 10.5%, protein 8.0%–11.5%, oil 4.5%, sugars 2.0%, and ash 1.5% [43,44,45]. Table 1 summarizes the chemical composition and physical properties of cornstarch. Corn hull fiber remains as residues of the starch isolation process [46]. Other types of corn fiber extracted via alkalization and other different treatments [47,48]. Table 2 shows the properties of corn stalk, hull, husk and cob. Corn husk and cob have the highest percentage of cellulose. Compared with other sources of fibers, such as wheat husk, rice husk, wheat straw, wheat bran and rice straw [49,50,51,52,53,54,55,56,57,58,59,60,61,62,63,64,65,66,67,68,69,70], corn fibers have the highest amount of cellulose.

**Table 1 polymers-14-04396-t001:** Physical properties and chemical composition of corn starch [46,71,72,73,74].

Contents	Amount	Units
Density	1.4029–1.356	g/cm³
Melting point	256°–258°	C
Boiling point	205°	C
Amylose	29.4–24.64	g/100 g
Amylopectin	75.36–72	g/100 g
Crude fats	7.13–0.32	g/100 g
Crude proteins	7.70–0.38	g/100 g
Ash	0.62–0.32	g/100g
Phosphor	0.09	%
Moistures content	10.82–10.45	%

**Table 2 polymers-14-04396-t002:** Physical properties and chemical composition of corn Stalk, hull, husk and cob [46,51,52,53,54,75,76,77,78,79,80,81,82,83].

Contents	Corn Stalk	Corn Hull	Corn Husk	Corn Cob
Density (g/cm³)	1.42	1.3231	1.49–1.18	0.8–1.2
Porosity (%)	58.51	-	88 ± 2	67.93
Cellulose (%)	32–39	15.3	31.3–47	40–44
Hemicellulose (%)	29.1–42	40.4	34–43.91	31–33
Lignin (%)	5–38.12	2.1–2.87	1.5–14.3	16–18
ASH (%)	24.9	0.88–1.3	3.3–6.8	2.3–3.2
Moisture (%)	3.32–6.4	4.2–8.59	7.6–8.7	6.38

## 4. Properties Improvement Techniques

There are two ways to improve biocomposites-based fiber properties, either using fiber surface treatment or additions such as grafting and cross-linking agent. While plasticizers are added to enhance the flexibility of biocomposites based starch matrix.

### 4.1. Plasticizers

Plasticizers are used in composite material to improve the flexibility and applicability of the biocomposite based starch. Starch cannot be melted in its native form due to the strong hydrogen bonds, plasticizer molecules penetrate starch granules and reduce hydrogen bonds in high temperature, pressure and shear stress [84].

It has been found that ethanolamine, diethanolamine (DEA), triethanolamine, ethylene glycol, and glycerol can act as plasticizers for starch [85]. Glycerol is widely employed to reduce the strong intermolecular hydrogen-bonding interactions of starch, decreasing their glass transition temperature [86]. In addition, it improves its processing property and increases the elongation at break [87]. However, its presence usually significantly leads to a decrease in the mechanical properties of the resultant materials [88,89]. 

Plasticizers have been used to promote plasticity and flexibility and reduce brittleness, but adding plasticizers must be experimented with to avoid the undesirable drop in the mechanical properties. For instance, reinforced plasticized corn starch with 8% of corn husk results in 12.9 MPa of tensile strength and 615 MPa of Young’s modulus. Adding 6% of sugar palm fiber (SPF) improves the mechanical properties, including tensile strength of 19 MPa and Young’s modulus of 1160 MPa. While the highest elongation was gained without reinforcing starch with fiber [90,91].

Corn starch can be plasticized by adding polyol mixtures of conventional plasticizer glycerol and higher molecular weight polyol (HP), such as Xylitol, Sorbitol, and Maltitol. When the glycerol is added to the corn starch, the highest tensile strength was 9.2 MPa, which is gained when the mixture contains 40 Parts per Hundred (PHR) of glycerol while the highest elongation at break was 117% at 60 PHR of glycerol. When sorbitol is added to corn starch, the highest tensile strength was 10.2 MPa gained when the mixture contains 50% of sorbitol of the polyol mixture while the highest elongation at break was 104% at 0–10% of sorbitol. The highest tensile strength gained was 16.26 MPa by the mixture of 20% Glycerol and Maltitol [92]. When plasticizer is added, hydrogen bonding is reduced, allowing molecules to move and increases elongation, whereas high tensile occurs when starch-starch hydrogen bonds overcome starch-plasticizer interactions in low amounts of plasticizer [93].

### 4.2. Surface Treatment

Surface treatment is a common method to clean, modify, or improve the natural fiber surface to lower surface tension and enhance interfacial adhesion between natural fiber and polymeric matrix [94,95]. The extra-cellulosic components that cover the fibrils are largely eliminated as a result of this treatment, contributing to a rougher surface topography [96]. Equation (1) shows the chemical reaction of alkaline treatment as an example [97]. The conception of surface treatment is to improve the adhesion property by break linkages of polyester backbone and generate hydrophilic hydroxyl and carboxylic groups [98]. This in turn improves morphological, mechanical, and thermal properties [99]. While the concept of cross-linker agent is a reaction. It is usually happening between the active groups of cross-linking agent, such as aldehyde groups, acid groups, and epoxy groups, with the amino groups of the material that the cross-linker added to it [100]. Several publications have addressed the effects of surface treatment on the structure and properties of natural fibers such as corn, kenaf, flax, jute, hemp, and sisal fiber [101,102,103,104].
(1)$$\mathrm{Fiber}-\mathrm{OH}\;+\;\mathrm{NaOH}\;=\;\mathrm{Fiber}-O^{-}\;+\;{\mathrm{Na}}^{+}\;+\;H_{2}O$$

There are many treatments available to improve the surface properties of natural fiber, some of these treatments are:-Acrylonitrile treatment improves the mechanical properties and decreased the maximum degradation temperature [105,106];-Maleic anhydride treatment improves compatibility between the constituents of biocomposite, enhance the mechanical properties and the melt temperature [85,107,108];-Titanate treatment improves the processibility of the biocomposite, and mechanical behavior [109];-Oxidative treatment increases the carbonyl and carboxyl and enzymatic hydrolysis of starch [110];-Alkali (sodium hydroxide) treatment for starch improved the pasting properties, reduced the past temperature and increased the peak viscosity [110];-Alkali and silane (as a coupling agent) treatment is one of the most effective surface treatment to remove out hemicellulose and lignin [111]. It is important to select the perfect amount of treatment solution, temperature, and the processing time.

### 4.3. Grafting

Bonding monomers onto the polymer chain improves the functional properties of the polymer such as sulfonation, phosphorylation, carboxymethylation and acetylation [112]. Grafting copolymerization in polymers is a method that can import the properties of polymer onto the composite, covalent bonds occur between the side chains and the fiber during grafting, resulting in a branched-chain structure [8,113]. It is a method to integrate and increase the interaction between properties of natural fiber and either natural or synthesis polymers matrix. Grafting method improves mechanical properties, thermal properties and water resistance, which increases the potential applications for natural fiber. Maleic anhydride, for example, is one of the agents utilized in grafting on cellulosic fibers. Graft copolymerization with maleate agents’ results in a chemical linkage coating on the fiber surface. This might reduce the hydrophilic characteristic of fiber, leading to improved interfacial bonding with polymeric matrices [114]. Grafting of natural fiber can divided in three groups: (1) grafting fiber with single monomer; (2) grafting fiber with two monomers; (3) grafting fiber with polymer [115].

Corn starch have been used as a matrix for grafted fiber such as *Saccharum spontaneum* L. (Ss) fiber and methyl methacrylates (MMA) [116,117]. Corn fiber is used to graft poly(dimethylaminoethyl methacrylate) (PDMAEMA) for adsorb Hexavalent chromium (Cr(VI)) in industrial application [118] and corn cob cellulose is used to graft polyacrylonitrile [119]. Corn fiber have been grafted with maleic anhydride and vinyl acetate. Figure 3 shows grafted cellulose with maleic anhydride and vinyl acetate [120,121,122].

Grafting Technique Types:

#### 4.3.1. Grafting by Living Polymerization

In this technique, polymer is grafted onto cellulose and the most popular methods in this technique are ring opening polymerization and free radical grafting.

##### Ring Opening Polymerization

Ring Opening Polymerization (ROP) is polymerization in which cyclic monomer yield a monomeric unit [123]. It changes the mechanism of polymerization to yield higher molecular weight in shorter time [124]. ROP leaves no residues in the product, which means there exist a good molecular distribution [125].

##### Free Radical Grafting

Free radical grafting makes formation in the polymer by the successive addition of free radical building blocks. This method is good for starch grafting. However, this method is a high cost method, which makes it inappropriate for many applications [126].

#### 4.3.2. Grafting by Coupling Agent

Grafting via coupling agent improves the adhesion between the fiber and the matrix such as isocyanates. N-Octadecyl isocyanate is used as a coupling agent to improve compatibility between the Nano-crystalline cellulose extracted from Sisal and poly(caprolactone) (PCL) matrix [127], while Methylenediphenyl diisocyanate (MDI) is used as a coupling agent for poly(lactic acid) (PLA) and starch [128].

#### 4.3.3. Ionic Grafting

Chemical grafting of the polymer can be achieved through ionic mode. Ionic grafting is similar to free radical generation method but in this method chemical initiator do not form free radicals. Instead, they form cationic or anionic centers that will initiate the grafting process [129]. The ionic liquids can be divided in three groups: cation, anion, and physical state at room temperature. 1-butyl-3-methylimidazolium chloride [BMIM]Cl and 1-allyl-3-methylimidazolium chloride [AMIM]Cl were the predominant Ionic solvents and reaction media for starches such as corn starch [130]. Corn starch was grafted in [AMIM]Cl and [BMIM]Cl, which enhances the gelatinized and catalyze the decomposition [131,132].

### 4.4. Cross-Linking Agents

Starch-based polymer blends are able to alleviate the environmental damage and alternate artificial plastics. However, many of starch-based polymer suffer from poor mechanical and thermal properties. Cross-linkage agents are used to improve the properties of biocomposites by improving the filler matrix interaction and filler dispersion. The most workable cross-linking agents are GLA and EP [19]. These two cross-linker improved the mechanical properties and thermal stability of chitosan/corn. There are other types of cross-linking agents that also illustrate good enhancement of other biocomposites. For example, in situ chemical cross-linking with Chitosan improves the capability of water resistance [100]. Aglycone geniposidic acid cross-linking with Chitosan film had higher tensile strength. It also exhibits slower degradation rate and lower water vapor permeability compared to Chitosan film without aglycone geniposidic acid [133].

## 5. Properties of Biocomposites Based Corn

### 5.1. Degradation

Biodegradation is biochemical process driven by material tendency and micro-organisms to revert to its original constituents. It is considered as one of the most important characters of biocomposite materials. The biodegradation behavior dictates if the material is suitable for application at environment. In addition, it provides information on the required disposal process. Examples of popular degradation tests are as follows:

#### 5.1.1. Thermal Degradation

Thermal degradation of polymer is the decay of molecules and loss of a hydrogen atoms from the polymer chain as a result of overheating. Thermal degradation serves as an upper bond to operating temperature of polymer. Thermogravimetric Analysis (TGA) is a technique to measure the degradation of the polymer by varying the heating rate while the change in the sample mass is measured. Differential Thermal Analysis (DTA) and Differential Scanning Calorimetry (DSC) is used to analyze the heating effect of polymer during the physical changes (glass transition and melting) [134,135]. Addition of plasticizers doesn’t significantly affect thermal properties of films based corn starch [136], while the addition of fiber improves the decomposition temperature of the film [46].

#### 5.1.2. Enzymatic Degradation

Polymeric materials depolymerize using micro-organisms’ secreted enzymes [137]. In this method, dry specimen is placed in enzymatic solution at 37 °C for a predetermined time. To maintain the enzyme activity at a desired level throughout the experiment, the enzymatic solution should be changed when the enzyme activity decrease. The time course of the weight loss and degradation behavior is evaluated using High Performance Liquid Chromatography (HPLC) [138,139].

#### 5.1.3. Soil Burial

Micro-organisms such as bacteria and fungi in the soil cause the degradation in the material after burial. This microorganism works by enzymatic action. The sample must be buried in soil with known weight for different time intervals, depending on the composite material and the application of the product. Then, the rate of degradation is evaluated by measuring the weighted and water absorption of the samples [139]. Reinforcing corn starch matrix with corn husk increased the degradable weight from the film [91]. The film with highest corn husk fiber contents shows higher decomposition resistance, making it more susceptible to mycobacterial attack. This microorganism is in the form of fungi and bacteria [140,141].

Cellulose is the most abundant polymer in corn fiber, while hemicellulose has the second percentage cellulose in the fiber composition. The strength of the chemical bonds in hemicellulose is lower than that in cellulose, therefore, hemicellulose starts to decompose first [142]. Lignin on the other hand, hinders the degradation of fiber structures which can be improved with some lignin decomposing enzymes [143].

### 5.2. Morphology Characterization

There are many techniques used to identify the morphology of materials such as Scanning Electron Microscopy (SEM), Transmission Electron Microscopy Analysis (TEM), Optical Light Microscopy (LM). Usually Scanning Electron Microscopy (SEM) is used identify the morphology of biocomposite materials.

SEM results illustrate that there is improvement in the morphology of the corn composite product after alkali treatment shows significant reduction in hemicellulose and lignin. Figure 4 shows corn fiber before and after alkali treatment while Figure 5 illustrates the enhancement that results from treating corn cob fiber with reinforced chitosan film.

Another way to improve its properties is to add cross-linkers. Adding cross linker to the combination of corn, cob and chitosan shows best in 20% of corn cob and 1% of the cross linker. The most efficient cross-linking agents are EP and adipic acid (ADP) which uses chitosan/corn cob biocomposite films [22]. The morphological analysis exhibited an improvement in the interfacial interaction and a good homogeneous between the composites when plasticizers were added. Adding thermoplastic corn starch to chitosan or chitin aminopolysaccharide polymer results to homogeneous and smooth surfaces, without pores and cracks and with a clear zone surrounding the circular film. Figure 6 shows the improvement in the corn starch films after the addition of chitosan [144].

Figure 7 shows the morphology of the corn husk, stalk, and cob fibers after isolation and bleaching. When cob fibers samples showed normal size, it was easy to see that the length and diameter of corn husk and stalk fibers were uneven. The average lengths of the husk and stalk fibers were 218.59 ± 165.36 and 218.45 ± 184.05 μm, respectively, while the diameters were 17.78 ± 15.59 and 18.32 ± 9.07 μm [145].

### 5.3. Crystallinity

X-ray diffraction (XRD) is an analytical technique used to characterize crystalline phases of a wide variety of materials, The advantages of using X-ray diffraction are its capability to characterize a single crystal with high-accuracy [146]. XRD test is used to study the crystallinity of the material. At the angular scattering of 2θ, the crystallinity index (*CI*) can be determined using Equation (2).
(2)$$CI=\left(\frac{A_{c}}{A_{c}+A_{a}}\right)\times 100$$
where *A_a_* is the amorphous area and *A_c_* is the crystalline area [136].

Chemical treatment, adding crosslinking agent and fiber grafting improves the crystallinity of corn fiber while plasticizers improve corn starch crystallinity. Adding 6% and 8% of palm fiber to corn husk as reinforcement for corn starch matrix reveals 23.9% and 27.3% of crystalline index, respectively [90]. Corn fiber treated with alkali, alkali and silane, silane, revealed 65.7%, 64.4% and 69.7% of crystalline index, respectively [147]. 8% of corn fiber reinforcing matrix of corn starch revealed 25.7% of crystalline index [91]. Corn Starch + 55% Fructose, Corn Starch + 55% Sorbitol and Corn Starch + 55% Urea, revealed 16.7%, 72.4%, and 35.5% of crystalline index, respectively [136]. Corn starch in different sizes of micro particles 300 μm, 300–150 μm, 150–74 μm and 74–37 μm < 15 μm revealed 58.14%, 57.37%, 60.42%, 62.52% and 64.27% of crystalline index, respectively [148].

### 5.4. Water Vapor Permeability (WVP)

To measure the passage of water vapor through a substance we need to calculate Water Vapor Transmission Rate (*WVTR*) or according to the ASTM E96-00 standard [16,149,150].

Water Vapor Permeability is a measure for the permeability of vapor barriers. Before starting the test of water vapor permeability, the films were kept under 25 °C and 67% relative humidity in a desiccator for 48 h, the specimen placed at the top of the test cup and sealed with melted paraffin. The cup prefill up with an hydrous calcium chloride, leaving 3 mm above to the top, after that the sample place under conditions 25 °C and 100% relative humidity (RH), according to stander ASTM E96-00 [16,149,150]. Water Vapor Permeability (*WVP*) calculated by the Equations (3) to (5):(3)$$WVP=\frac{m\times d}{A\times t\times P}$$
(4)$$WVTR=\frac{m}{A\times t}$$
(5)$$WVP=\frac{WVTR}{P\left(R_{1}-R_{2}\right)}\times d$$

Referring to Equations (2)–(4), *d* is film thickness, *m* is the weight increment of the cup, *A* is the area exposed, *t* is the time lag for permeation, *P* is water vapor partial pressure difference across the film. *R*_1_ is the relative humidity RH in the desiccator, *R*_2_ is the relative humidity RH in the cup and [151].

The water vapor permeability (*WVP*) of corn starch usually ranges between 7–8 g.mm m^−2^ d^−1^ kPa^−1^ [152]. Filling Pectin-based composite with corn husk fiber reduces the water vapor permeability from 11 g/m.s.Pa (at 0% *w*/*w* concentration of corn husk) to 8.5 g/m.s.Pa (at 5% *w*/*w* concentration of corn husk) [152]. Using glycerol to plasticized films of chitosan and corn starch reduces water vapor permeability form 8.8 ± 0.9 * 10^−11^ g s^−1^ m^−1^ Pa^−1^ at (without glycerol) to 4.5 ± 0.4 * 10^−11^ g s^−1^ m^−1^ Pa^−1^ (using glycerol as plasticizer) [153]. It’s important to reduce the water vapor permeability (*WVP*) to minimize moisture transfer in the final product.

### 5.5. Oxygen Permeability

Oxygen permeability can be used to define how easily oxygen passes through a particular material. It can be measured using gas permeation instrument using ASTM 3985 standard. Permeability of one layer can be determined using multilayer permeability Equation (6):(6)$$\frac{L}{P}=\overset{n}{\underset{i=1}{\sum }}\frac{L_{i}}{P_{i}}$$
where *n* is the number of layers, *L* and *P* are the thickness and permeability of the multilayer film, *L_i_* and *P_i_* are the thickness and permeability of each layer, respectively [154].

The oxygen permeability value of the pure starch film was 12.11 cm³ µm.m^−2^ d^−1^ kPa^−1^ [152]. It has been observed that when the water content in starch is low, the oxygen barriers is good and when the water content increases the oxygen barriers do not exist [155]. Plasticizing corn–zein coating structure on polypropylene (PP) with glycerol (50% concentration) and polyethylene glycol (PEG) (20% concentration) reduced oxygen permeability to 44.0 cc/m² Χ day (5% corn–zein (*w*/*v*) concentration), 41.5 cc/m² Χ day (15% corn–zein (*w*/*v*) concentration), respectively [156].

### 5.6. Chemical Characterization

To identify chemical bonds in organic materials, polymers, metals and various sorts of materials, we used Fourier Transform Infrared Spectroscopy (FTIR) [157]. The FTIR analysis method uses infrared light to scan test samples and observe chemical properties [158]. When a material is irradiated with infrared radiation, absorbed IR radiation usually excites molecules into a higher vibrational state. The wavelength of light absorbed by a particular molecule is a function of the energy difference between the at-rest and excited vibrational states. The wavelengths that are absorbed by the sample are the characteristic of its molecular structure.

The FTIR result shows that the lignin and most of the hemicellulose were removed from the corn fiber using chemical treatment [94,95]. Cross-linking agents results to increased C = N (Imine linkages), C = C (Ethylenic bonds), C-O-C stretching and C-H stretching (Aromatic ring of SAL) [5] while corn treated by alkali treatment shows a decrease and stretching in C = O bonds, which corresponds to ester bonding of hemicellulose [159]. Treating corn fiber with alkali also shows stretch in C = C bonding which is corresponding to reduction of hemicellulose [147].

### 5.7. Thermal Properties

Thermal stability of biocomposite material is evaluated by monitoring change in weight as a function of temperature at a predetermined heating rate. Thermal properties of biocomposite materials are usually measured by Thermogravimetric analysis (TGA), Differential Scanning Calorimetry (DSC), Thermomechanical Analysis (TMA), and Dynamic Mechanical Analysis DMA [160]. Thermal results revealed that the corn starch based biocomposite degrades at 161.2 °C, while hull, husk and stalk fiber began to degrade above 260 °C. This makes corn fiber a good composite material because the majority of composites are processed at above 180 °C [46]. The low degradation temperature of films based starches can be improved by reinforcing film based starch with fiber, 2%, 4%, 6%, and 8% of corn starch with corn husk revealed 279.8 °C 280.5 °C, 280.53 °C 278.63 °C for maximum temperature, respectively [46]. Using cross linker also improves the degradation temperature 20% of corn cob with chitosan with 1% of EP cross-linker degrade at 290 °C [22]. The alkali treatment for corn fiber keeps constant the maximum temperature, which gives the opportunity to add more fiber to the composite. LDPE/alkali treated corn fiber of the following load 0%, 10%, 30% and 50% revealed 112.7 °C, 112.2 °C, 112.6 °C and 112.7 °C maximum temperature, respectively [161].

Corn starch plasticized with citric acid and reinforced with Grewia optiva fiber and methyl methacrylate (MMA) grafted fibers in addition of GLA as a cross-linker improves the thermal stability and the decomposition temperature can reach up to 522.65 °C [162].

At the onset temperature stage, which is approximately 300 °C for films based starch, the starch carbon chains polymer underwent hydrogen functional group removal, degradation, and depolymerization [163]. By adding additives, such as fiber and cross-linkers, to create strong connections, you can postpone temperature deterioration [164].

### 5.8. Mechanical Characterization

Mechanical characterization can be carried out using standard tests. The ASTM standard are summarized in Table 3. Tensile strength, elongation, flexural strength, tensile modulus, and flexural modulus are the usual mechanical properties acquired.

Using grafting modification enhanced the mechanical properties of corn composite whereby the addition of maleic anhydride polyethylene (MAPE) to the composite of LDPE/CS improved the interaction between the LDPE/CS composite, at 10% of filler loading, tensile strength of the biocomposite was 8.1 MPa without grafting, and it increased to 9.5 MPa. However, the addition of MAOE decreased the elongation and young modulus [168]. Reinforcing corn starch with polypropylene coupled by maleic anhydride increased the tensile strength and reduced tensile modulus while there was no significant change in elongation [169].

Generally, mechanical properties can be improved by fiber surface treatments, addition of plasticizers and cross-linking agents [170,171,172,173]. Most researchers focus on mechanical properties as part of their investigation. Table 4 shows the mechanical properties behavior of corn based biocomposites.

### 5.9. Water Uptake

Water uptake or water absorption refers to the ability of a material to absorb water. The increase of water absorbed by material causes an increase in the swelling and decrease in the mechanical properties of the material. Water absorption is expressed as increase in weight of the material due to immersion of water in percentage. To measure the water uptake of the material, the specimen is soaked in water for a known time at a specified temperature. Then, the water uptake is calculated using Equation (7) [179]. Biocomposites are easy to absorb water and moisture because they have more holes or pores compared to synthetic composites. Corn fiber based composites revealed low water absorption which is reflected as improvement in mechanical properties [40].
(7)$$Water\;uptake\;\left(\%\right)=\frac{W_{t}-W_{0}}{W_{0}}\times 100$$
where, *W_0_* is the weight of the sample before drying and *W_t_* is the weight of the dried sample after *t* time.

Reinforcing corn starch with rice husk/walnut shell reduced the water absorption to 4.32% [180] while reinforcing corn starch with Barley Straw Particles increased water absorption [181]. Grafting corn starch with poly (methyl methacrylate) (PMMA) and then compounding with styrene-butadiene rubber (SBR) reduced water absorption at 30 parts per hundreds of corn starch. Water absorption is influenced by the immersion time in water [182]. Films based starches shows high amount of water uptake, however, the films made of smaller particles of starch shows higher water uptake [183].

## 6. Corn Biocomposite Applications

The renewability of agricultural crops, such as corn, is one of the significant advantages of biocomposites-based agriculture resources. However, this advantage is restricted in biocomposites based forest plants unless the green cover of the forest is continually replenished and renewed.

Corn-derived fibers and straw may be used to make several types of paper and paperboards. Because lignin has a high energy content and fuel value, it is widely utilized as a burning fuel. Additionally, lignin is utilized to make adhesives and binders, although these usages offer little value [184]. Daily use products, such as paper padding, foam cushioning, packaging pads, air cushioning, loose fill, and mailer bags, have been produced from corn based biocomposite.

Agro-industrial waste such as corn residues are difficult to recycle or dispose of, making them a serious environmental issue [145]. These wastes, on the other hand, are renewable, biodegradable resources with the potential to manufacture high-value items, particularly non-wood vegetable fibers [185,186,187]. Micro and nano cellulose that is derived from corn fiber and other fibers sources has great potential for many applications. Cellulose that has been nano- or micro-fibrillated is utilized as an additive in the papermaking process, coatings, medicinal, pharmaceutical, cosmetic, sanitary, flexible electronics, barrier materials, high-temperature thermal insulation, and chronic wound healing [188,189,190,191,192]. Nanocrystalline Cellulose (NCC) is useful in medical applications because of its small diameter, particularly in vascular grafts, electronics, catalysis, packaging applications, synthetic plastics or polymers, fuel cells, filtration, catalysis, tissue engineering, solar cells, and lithium-ion batteries [193,194,195,196,197,198]. Because of their excellent thermal stability and tensile strength, NCC bionanocomposites can also be used in fire extinguishers and automobile components [199,200,201,202,203]. Abrasive in cosmetics, absorbent, anti-caking, bulking, and aqueous viscosity raising agents, binder, emulsion stabilizer, slip modifier, and texturizer are just a few of the usages of microcrystalline cellulose [182,183].

As the starch is the major component of the corn kernel, accounting for up to 72% to 73% of its weight [204], it has high potential ability to be used in packaging applications [205]. Corn starch based film are colorless, odorless, non-toxic, and semi-permeable to oxygen and carbon dioxide. The addition of different additives such as plasticizers, fillers and cross-linkers improves the ability of corn starch based film to be used in various application.

## 7. Economic Value, Challenges, and Future Perspective for Corn-Based Composites

Production of bioplastics ensures the proper usage of biomass, whether its waste, such as corn residues, or prepared specially for bioplastic production, such as corn starch. Over the forecast period, 2017–2030, demand for bioplastics is expected to expand at a compound annual growth rate of 11.2% [206]. The increasing in demand of bioplastic show the important of developing models of production, consumption and at the same time it must be unharmful to the environment (circular economy) [207,208], using circular economy in biocomposite production will improve their recyclability, reusability, and ability to be transformed to a valuable goods [209]. However, the inherent bonding between biomatrix and filler, lack in waste management techniques, high energy that can be consumed in recycling operations, carbon footprints, and release of harmful chemical, all these issues can create a real challenge for the recirculation process [210].

Biodegradation is normally not aimed at recovering plastic materials or monomers to be reintroduced in the life cycle of plastic products, while this is specifically the aim of other types of recycling options, such as mechanical and chemical recycling, which address both waste management and primary resource preservation. However, to control material waste and environmental footprints, the utilization of novel biocomposite elements should be enhanced along with and development of innovative recycling technologies [207]. However, it has been found that thermoplastic polymer (TPS), such as that based on corn starch, could be reprocessed at least four times, without the need for adding virgin material [211].

According to a market study, global biocomposite usage reached $19.6 billion in 2020, with a projected increase to $38.07 billion and a compound annual growth rate (CAGR) of 14.2 percent in 2025 [210]. Meanwhile, the worldwide synthetic fiber industry, which was worth $147.16 billion in 2019, is only predicted to increase at a 5% CAGR by 2025, indicating a drop in consumption. Furthermore, global plant fiber output has risen at an exponential rate over the last two decades, from 107 million tons in 2018 to 145 million tons by 2030 [212].

Green polymers reinforced with natural fillers offer an exciting prospect for future research in order to reduce the usage of petroleum-based plastic. Recent and future studies focus on selecting the best biodegradable components of the biopolymer (matrix, filler, plasticizer, etc.) and optimizing processing parameters based on selected properties. Natural composites will become more affordable as a result of diligent study, while their quality will improve, markets will expand, and the environmental impact will be reduced [213].

## 8. Conclusions

Biocomposite material has great potential to replace synthetic plastic materials in various applications. Studies have been done to improve the properties of biocomposites. Corn fiber and corn starch emerged as popular biocomposites materials in the biocomposite research field. It has been found that both corn starch and corn fiber need to improve their properties. Adding cross-linking agents to the corn fiber, such as EP and GLA, and performing treatments to the corn fiber, such as using alkali and silane, maleic anhydride, and vinyl acetate to graft corn fiber has revealed improvement in thermal and mechanical properties of the composite. Adding plasticizers to corn starch improves the flexibility and applicability of the biocomposite material. Perfect composition percentages are the critical point for an ideal composite material with high properties. From an industrial perspective, bioplastics derived from starch may be processed at least four times without the addition of new material. However, further research must be done on the ways of managing and recycling biocomposite waste. Through plasticizing, chemical treatment, grafting, and cross-linker agent operations, the mechanical, thermal, and water resistance properties of corn starch and fiber based biopolymers are enhanced, which increases their potential uses.

## Figures and Tables

**Figure 1 polymers-14-04396-f001:**
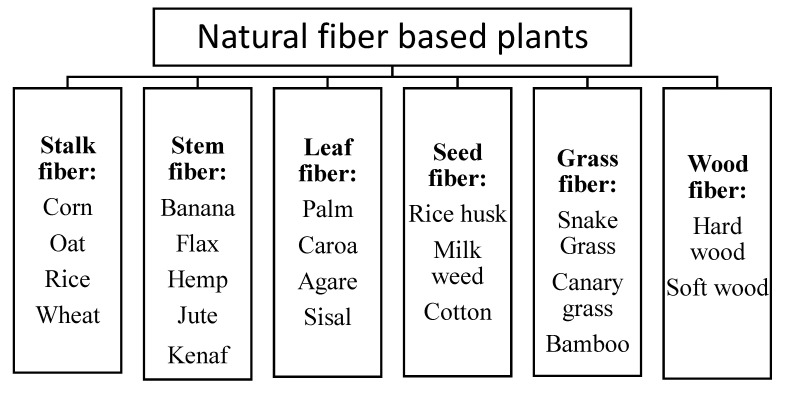
Classification of natural fiber [13,24,28].

**Figure 2 polymers-14-04396-f002:**
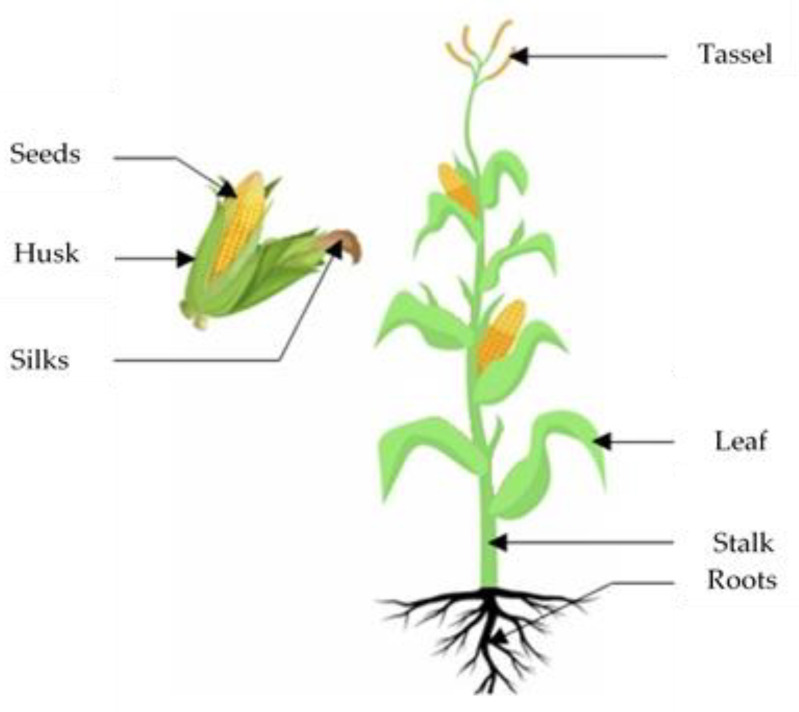
Different parts of corn plant.

**Figure 3 polymers-14-04396-f003:**
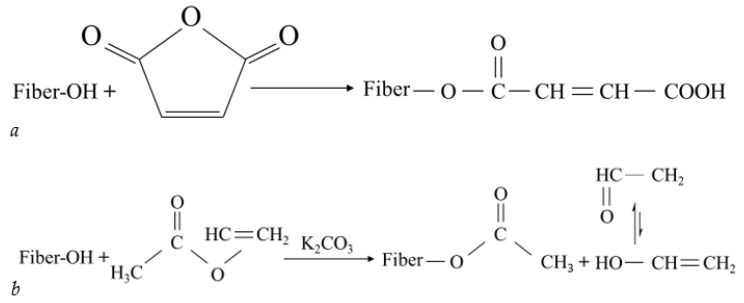
Grafted cellulose with (**a**) maleic anhydride and (**b**) vinyl acetate.

**Figure 4 polymers-14-04396-f004:**
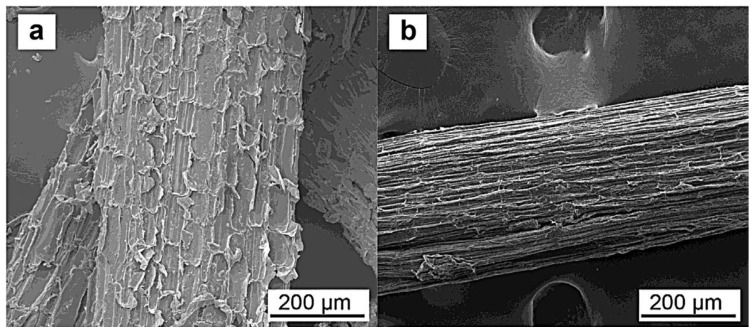
(**a**)Untreated corn fiber; (**b**) Alkali treated fiber. Reprinted with permission from Ref. [111], copyright 2022, Elsevier.

**Figure 5 polymers-14-04396-f005:**
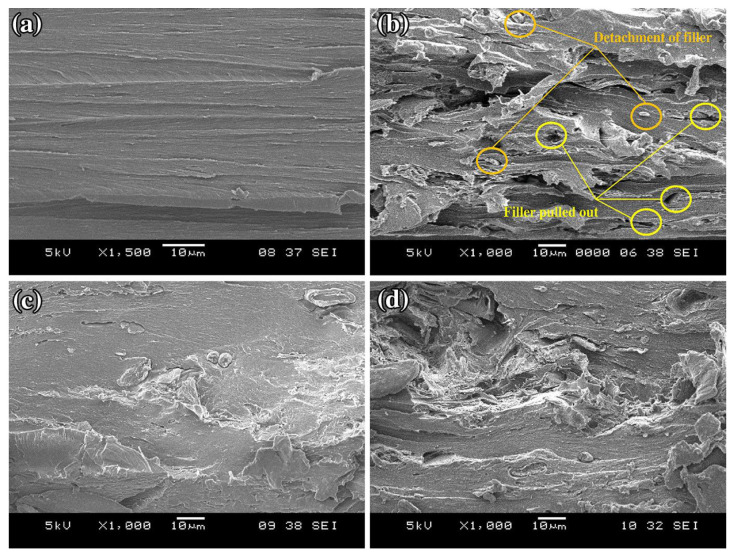
(**a**) SEM micrograph of neat CS; (**b**) SEM micrograph of unmodified CS/CC biocomposite film at 20 wt % of CC content; (**c**) SEM micrograph of modified CS/CC biocomposite film with ADP at 20 wt % of CC content; (**d**) SEM micrograph of modified CS/CC biocomposite film with EP at 20 wt % of CC content. Reprinted with permission from Ref. [22], copyright 2022, Springer Nature.

**Figure 6 polymers-14-04396-f006:**
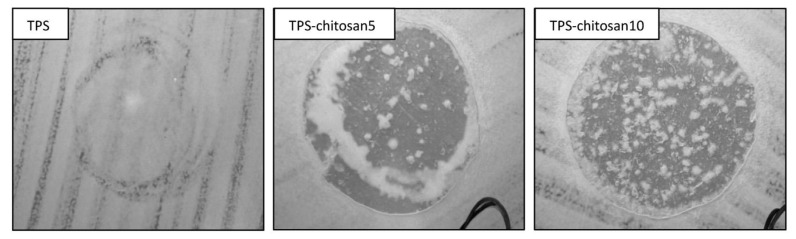
Improvement in the corn starch films morphology after adding chitosan. Reprinted with permission from Ref [145], copyright 2022, Springer Nature.

**Figure 7 polymers-14-04396-f007:**
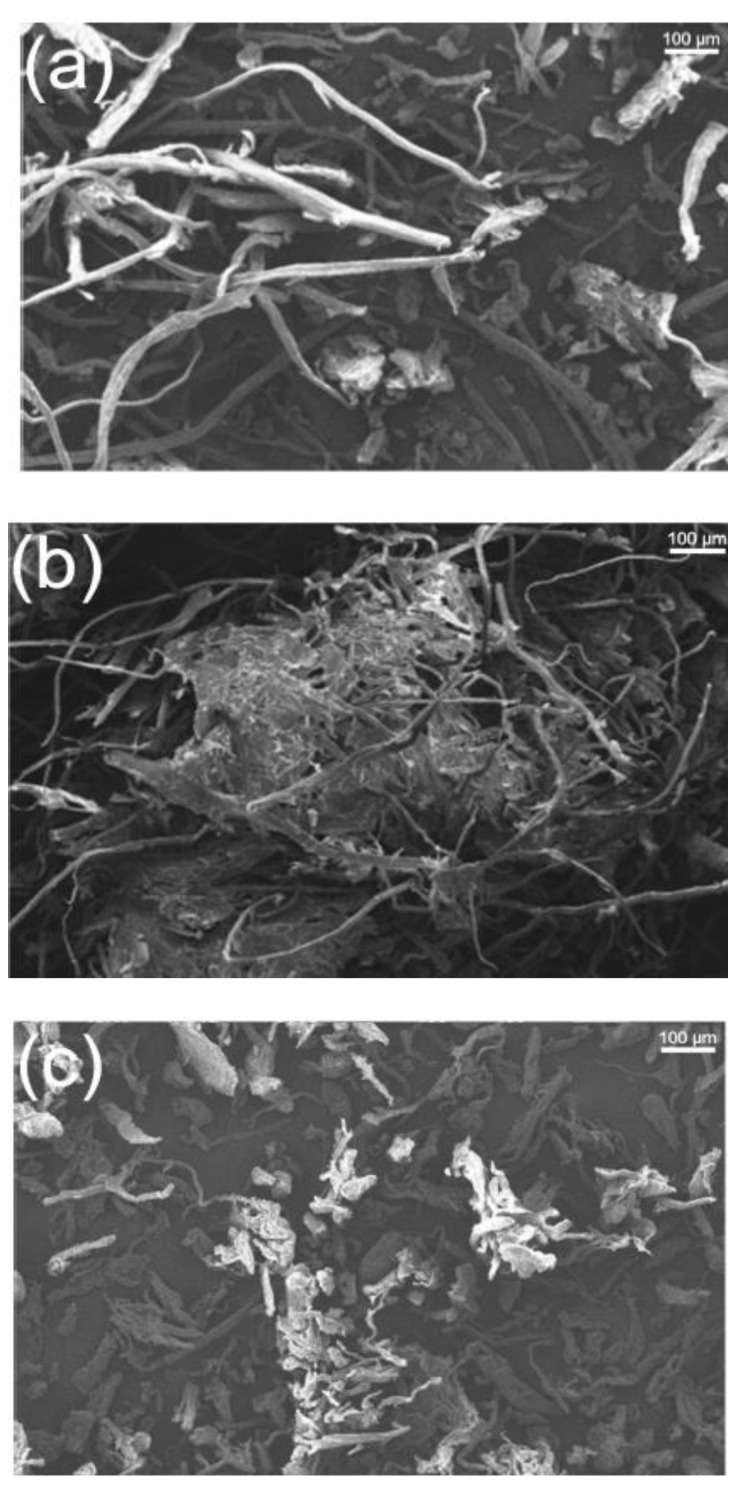
Morphology of isolated fibers from (**a**) corn husk, (**b**), stalk and (**c**) cob. Reprinted with permission from Ref [144], copyright 2022, Elsevier.

**Table 3 polymers-14-04396-t003:** Mechanical tests standards and the testing machines.

Testing Standards	Testing Machine	References
ASTM D638-91	Universal Testing Machine LK10k (Hants, UK)	[15,165]
ASTM D828–97	A TA. XT Plus Texture Analyzer	[16]
ASTM standard method D882-09 18	A Mecmesin MultiTest 1-í universal test machine	[17]
ASTM D 638-10	Universal Testing Machine (Zwick Co., Ulm, Germany)	[18]
ASTM D 882	Instron Universal Tensile Machine	[21]
ASTM D3039-76	Universal Testing Machine INSTRON H10KS	[166]
ASTM D 638 M-91a	TA.XT Plus Machine (SMS, Surrey, UK)	[167]

**Table 4 polymers-14-04396-t004:** Mechanical properties of corn based biocomposites.

Fiber Reinforce Material	Matrix	Plasticizer	Plasticizer %	Reinforcing Material %	Tensile Strength (MPa)	Young’s Modulus (MPa)	Flexural Strength (MPa)	Elongation (MPa)	Ref
-	Corn starch	Glycerol	10% 20% 30% 40%	-	60 30 17 9	62 32 20 12	-	10 25 45 55	[174]
Graphene oxide (GO)	Corn starch+ soy protein isolate (SPI)	Glycerol	30 wt% 30 wt% 30 wt% 30 wt% 30 wt%	GO 0 GO 0.5 GO 1 GO 1.5 GO 2	6.65 ± 0.9 2.84 ± 0.48 2.32 ± 0.35 3.60 ± 0.6 5.9 ± 0.27	252 ± 5.2 236 ± 48 332 ± 26 359 ± 25 449 ± 28	-	7.25 ± 43 19.73 ± 3.38 17.9 ± 3.4 10 ± 2.02 11.83 ± 2.08	[175]
Graphene (G)	Corn starch+ soy protein isolate (SPI)	Glycerol	30 wt% 30 wt% 30 wt% 30 wt% 30 wt%	G 0 G 0.5 G 1.0 G 1.5 G 2.0	6.65 ± 0.9 16.53 ± 0.26 11.38 ± 1.4 8.09 ± 1.4 6.66 ± 1.3	252 ± 502 578 ± 105 380 ± 48 329 ± 34 132 ± 55	-	7.25 ± 0.43 2.7 ± 0.12 1.2 ± 0.21 2.83 ± 1.83 6.58 ± 0.98	[175]
Corn cob	Low Density Polyethylene	-	-	0 40 45 50 55 60	14.9 13.8 13.0 15.8 11.3 9.4	130 133 138 141 148 163	-	11.3 10.2 9.2 7.8 7.1 5.8	[176]
Corn husk	Epoxy	-	-	0 5 10 15 20 25 30	34 29.2 26.2 22.5 19.4 16.2 14.8	-	48 40.8 33.7 29.3 25.6 23.5 21.8	-	[166]
Corn flour	poly (butylene succinate-co-butylene adipate) blends	-	-	70 50 30 0	3.93 ± 0.58 7.33 ± 0.96 11.9 ± 0.6 50 ± 0.2	189 ± 15 235 ± 11 279 ± 10 265 ± 5	-	16 ± 2 38 ± 14 64 ± 36 800 ± 10	[177]
corn cob, EP, ADP, GLA	Chitosan (CS)	-	-	0 20 20 + EP 20 + ADP 20 + GLA	52.8 ± 2.3 33.1 ± 1.7 46.9 ± 0.9 33.71 ± 2.01 42.0 ± 1.6	2269 ± 186 2571 ± 121 2703 ± 177 2616 ± 210 3178 ± 174	-	13.8 ± 0.4 8.7 ± 0.7 8.2 ± 0.4 12.18 ± 0.3 7.7 ± 0.2	[22]
Corn cob Treated by acrylic acid	Soy Protein Isolated	-	-	0 10 20 30 40	2.6 3.6 4.5 4.8 5.7	35 65 130 140 195	-	49 34 19 12 8.5	[178]
Corn husk fiber, Alkali(A) Alkali and silane(AS) treatment	Polylactic acid	-	-	0 5 A 10 A 15 A 10 A-1S 10 A-2S 10 A-3S 10A 10-4S	47 56 64 61 60 69 67.5 66	-	80.5 89 109 102 103 113 114 106	-	[159]
Corn husk	Corn starch	Fructose	25% 25% 25% 25%	2 4 6 8	8.8 7.8 10.9 12.9	320 190 510 615	-	-	[91]
Corn husk (CH), sugar palm fiber (SPF)	Corn starch	Fructose	25% 25% 25% 25%	8%CH + 2%SPF 8%CH +4%SPF 8%CH + 6%SPF 8%CH + 8%SPF	17 17.3 19 18	1050 1100 1160 1110	-	1.3 1.15 0.9 1.1	[90]

## Data Availability

No data were used to support this study.

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
