# Peer review of "Corn: Its Structure, Polymer, Fiber, Composite, Properties, and Applications"

_polymers, 2022, doi:10.3390/polym14204396_

Round 1

Reviewer 1 Report

The writing of this manuscript should be improved. Besides, you can find my comments below.

1. Lines 46-48: There is many research are conducting on wood and non-wood plants to extract fiber for biocomposite materials.

This sentence is confusing.

 2. Line 48: … extracted from varies types of …

varies or various?

 3. Lines 63-64: … like alkali, saline, alkali, compatibilizer and 64 adding cross-linking agents such as glutaraldehyde (GLA) and epichlorohydrin (EP) …

Alkali appears twice? Why?

 4. Figure 1: How many corns can grow in one plant?

 5. Line 148: The corn grain composition is; starch 68.0%-74.0 %, …

Why is there a semicolon here?

 6. Line 168: …, plasticizers molecules penetrate …

plasticizers → plasticizer.

 7. Figure 3: Is it methyl or methylene structure in the product? Please confirm the chemical structure.

8. How about the recyclability of the corn-based materials?

Author Response

Response to Reviewer 1 Comments

Point 1: Lines 46-48: There is much research are conducting on wood and non-wood plants to extract fiber for biocomposite materials.

This sentence is confusing.

Response 1:

Thank you for this point, we replaced the sentence in the same lines with “Numerous investigations are being conducted on plants, both wood and non-wood, to extract fibers for bio-composite materials.”

Point 2: Line 48: … extracted from varies types of …

varies or various?

Response 2:

Thank you for this important point. Word has been corrected.

Point 3: Lines 63-64: … like alkali, saline, alkali, compatibilizer and 64 adding cross-linking agents such as glutaraldehyde (GLA) and epichlorohydrin (EP) … 

Alkali appears twice? Why?

Response 3:

Thank you for this valuable comment, we corrected the sentence to alkali, saline, alkali- saline, as saline and alkali treatments have been use separately and in some cases, they have been used together.

Point 4: Figure 1: How many corns can grow in one plant?

Response 4:

Thank you for this valuable comment. Maximum 2 corns (ears) in each stalk, therefore, we remove three ears in figure two and we kept two to indicate this point.

Point 5: Line 148: The corn grain composition is; starch 68.0%-74.0 %, …

Why is there a semicolon here?

Response 5: Thank you for this valuable point, the semicolon has been replaced with “consist of”, so the sentence has changed to “The corn grain composition consists of starch 68.0%-74.0 %...”.

Point 6: Line 168: …, plasticizers molecules penetrate …

plasticizers plasticizer.

Response 6:

Thank you for highlighting this point, the word plasticizers have been changed to plasticizer.

 Point 7: Figure 3: Is it methyl or methylene structure in the product? Please confirm the chemical structure.

Response 7:

Thank you for highlighting this point, grafting fiber with maleic anhydride is associate with the methyl and methylene groups, while grafting fiber with vinyl acetate is associate with methyl groups, we added references 121 and 122, besides 120, those three references includes and confirm this information.

Point 8: How about the recyclability of the corn-based materials?

Response 8:

Thank you for highlighting this point, we added new paragraph from line 631-639; “Biodegradation is normally not aimed at recovering plastic materials or monomers to be reintroduced in the life cycle of plastic products, while this is specifically the aim of other types of recycling options, such as mechanical and chemical recycling, which address both waste management and primary resource preservation. But to control material waste and environmental footprints, the utilization of novel biocomposite elements should be enhanced along with and development of innovative recycling technologies [207]. However, it has been found that thermoplastic polymer (TPS) such as the one based corn starch could be reprocessed at least four times, without the need of adding a virgin material [211].”.

Reviewer 2 Report

comments

1- Brief results should be added to the abstract.

2- The figures, References, and tables should rearrange according to MDPI style.

3- The conclusion should be improved. 

Author Response

Response to Reviewer 2 Comments

Point 1: Brief results should be added to the abstract.

Response 1:

Thank you for this point, we added those sentences in lines 33-36 as a brief result: “The mechanical, thermal, and water resistance properties of corn starch and fibers based biopolymers show a significant improvement through plasticizing, chemical treatment, grafting, and cross-linker agent procedures, which expands their potential applications.”

Point 2: The figures, References, and tables should rearrange according to MDPI style.

Response 2:

Thank you for this important point. Figures, references, and tables have been rearranged according to MDPI style.

Point 3: The conclusion should be improved.

Response 3:

Thank you for this valuable comment and advice, we did a grammatical correction besides expanding the conclusion by adding those sentences in lines 671-676: “From an industrial perspective, bioplastics derived from starch may be processed at least four times without the addition of new material. However, further research must be done on the ways of managing and recycling biocomposite waste. Through plasticizing, chemical treatment, grafting, and cross-linker agent operations, the mechanical, thermal, and water resistance properties of corn starch and fibers based biopolymers are enhanced, which increases their potential uses.”.

Reviewer 3 Report

In this paper, the authors conducted a comprehensive review on corn fiber and corn starch based on composites. The review started with the advantages of using bio-based composites instead of artificial plastic materials to reduce environmental pollution, then introduced the history of corn, the composition of corn plant structure, and ways for starch extraction and preparation. Then the authors reviewed the properties of the corn based bio-composites (degradation, morphology, crystallinity, thermal properties, and chemical characterization) and techniques to improve the properties of bio-based composites. Finally, the application and challenges of corn-based composites were discussed.

The overall aspect of this paper is good. It is a work of certain interest for readers of Polymers.

Minor comments:

I suggest the authors to replace the "bio-composites based corn" with "Corn based bio-composites" in this review.

Author Response

Response to Reviewer 3 Comments

Point 1: I suggest the authors to replace the "bio-composites based corn" with "Corn based bio-composites" in this review.

Response 1:

Thank you for this point, we replaced bio-composites based corn with Corn based bio-composites in lines 502 and 517.